# Host Gene Regulation by Transposable Elements: The New, the Old and the Ugly

**DOI:** 10.3390/v12101089

**Published:** 2020-09-26

**Authors:** Rocio Enriquez-Gasca, Poppy A. Gould, Helen M. Rowe

**Affiliations:** Centre for Immunobiology, Blizard Institute, Queen Mary University of London, London E1 2AT, UK; r.enriquez-gasca@qmul.ac.uk (R.E.-G.); p.gould@qmul.ac.uk (P.A.G.)

**Keywords:** gene regulation, transposable elements, endogenous retroviruses, epigenetic repression, Intracisternal A-type particle elements, position-effect variegation, KRAB-associated protein 1, X chromosome inactivation, genomic imprinting

## Abstract

The human genome has been under selective pressure to evolve in response to emerging pathogens and other environmental challenges. Genome evolution includes the acquisition of new genes or new isoforms of genes and changes to gene expression patterns. One source of genome innovation is from transposable elements (TEs), which carry their own promoters, enhancers and open reading frames and can act as ‘controlling elements’ for our own genes. TEs include LINE-1 elements, which can retrotranspose intracellularly and endogenous retroviruses (ERVs) that represent remnants of past retroviral germline infections. Although once pathogens, ERVs also represent an enticing source of incoming genetic material that the host can then repurpose. ERVs and other TEs have coevolved with host genes for millions of years, which has allowed them to become embedded within essential gene expression programmes. Intriguingly, these host genes are often subject to the same epigenetic control mechanisms that evolved to combat the TEs that now regulate them. Here, we illustrate the breadth of host gene regulation through TEs by focusing on examples of young (The New), ancient (The Old), and disease-causing (The Ugly) TE integrants.

## 1. Introduction

In contrast to their paramount functional importance, protein-coding genes constitute only a small fraction (~2–4%) of the total DNA sequence of the human genome. Exquisitely regulated control of coding genes in time and space is a defining feature of development of multi-cellular organisms. For example, transcription can be regulated by the generation of multiple isoforms of the same gene by alternative splicing, alternative promoter/enhancer usage, non-coding RNAs and epigenetic modifications, which control chromatin structure and function (reviewed in [1]). On the other hand transposable elements (TEs) constitute an estimated two thirds of the human genome [2,3], and contribute to the regulation of protein-coding genes through their regulatory elements. TEs exercise a complex dialog with their host genomes that is distinct from a conventional virus-host arms race because they are not only potential parasites, but also a vital source of genome innovation [4,5,6,7,8]. TEs are subject to epigenetic silencing by histone modifications and DNA methylation [9,10,11] and become mutated and inactive over the course of evolution. A fraction, however, are co-opted and preserved under purifying selection.

In this review, we illustrate how TEs have been co-opted to regulate host genes by focusing on TEs that can alter their surrounding epigenetic context, with our goal to highlight TEs as a normal feature of host gene regulation. The fact that TEs are so ubiquitous in the genome, contain their own regulatory sequences and have become hotbeds of epigenetic regulatory marks, due to their initial transcriptional silencing, means that they are ideally placed to re-shape host gene expression profiles. We will journey back in time to explore first how young or ‘new’ TEs, followed by ‘old’ TEs regulate mammalian genes. New TEs are here defined as specific to the primate or murine lineage, whereas old TEs correspond to those which predate the split between mouse and human ancestral lineages (see Figure 1 for examples selected in this review). This distinction allows us to emphasize that, while gene regulatory mechanisms involving TEs are generally conserved across species, the precise TEs that rewire genes are often species-specific. This is due to the different TE invasions that each species has encountered. We finally review instances whereby ‘ugly’ TEs have been retained by the host genome, likely due to them being beneficial, as well as potentially detrimental and therefore discuss the risk that TE co-option poses.

## 2. Gene Regulation by Transposable Elements: The New

Since the human and mouse lineages diverged from a common ancestor around 80 million years ago, their genomes have been subject to different selective pressures, innovations and invasions. The present-day human genome has been found to contain no endogenous retroviruses (ERVs) capable of replication/transposition [12], but to host around 100 retrotransposition competent Long INterspersed Element 1s (LINE-1s or L1s) [13]. The human genome is also home to SVA elements, a newly evolved composite TE derived from SINEs and an ERV (HERV-K10). SVAs harbour a variable number of tandem repeats (VNTRs) and hijack L1 retrotransposition machinery for their mobilisation. With the youngest SVA family (SVA_F), around three million years old (myo), SVA elements represent the youngest TE in the human genome [14]. In contrast, the mouse genome appears to contain cohorts of ERVs and L1s capable of retrotransposition [15,16]. Here, we inspect examples of epigenetic control of host genes through regulatory sequences embedded in young species-specific TEs. We draw on mouse and human examples and discuss how this can shape our understanding of how TEs underpin human adaptation and genetic variation. We include scenarios whereby parallel TEs have been independently co-opted for the same purpose in both organisms. Future studies on actively transposing TEs may allow us to observe how genome invaders become co-opted into gene-regulatory networks in real time.

### 2.1. New Transposable Elements in Mouse

Mouse-specific endogenous retroviruses include the Intracisternal A-type particles (IAPs), which adapted to retrotranspose intracellularly following loss of their envelope gene [17] and murine endogenous retrovirus L (MERVL). A small fraction of IAP elements can still retrotranspose and this subfamily has been a source of polymorphisms that have been actively studied for their effects on the expression of nearby genes [18,19,20]. In this section, we will focus on examples of gene regulation through specific MERVL and IAP-derived regulatory elements, which provides insight into how ERVs may directly influence gene expression and host fitness. Mechanisms by which ERVs that are discussed in this review regulate host genes are summarized in Figure 2.

#### 2.1.1. Co-Option of TEs to Regulate Gene Networks

Perhaps one of the best examples of a co-opted TE regulating a network of genes is MERVL LTR (MT2) promoters driving expression of genes specific to the totipotent 2-cell (2C) stage of development [25,26]. It is possible that this TE-gene network evolved due to MERVL invasions into genes actively expressed during this stage of development, which represents a window of opportunity for escape from ERV repression due to epigenetic reprogramming. Alternatively, insertion prior to differentiation of the germline would also represent a selective advantage to the TE [27]. Originally identified as TEs, which generate chimeric transcripts with host genes in cleavage-stage mouse embryos [28], MERVL expression was later shown to be associated with enhanced developmental potency in in vitro and in vivo assays [23]. Further work has shown that the MERVL-2C gene network is activated following depletion of the chromatin assembly factor-1, CAF-1, in mouse embryonic stem cells (mESCs), through increased chromatin mobility [29] and is associated with genome-wide DNA demethylation, potentially through upregulation of the translation inhibitor *Eif1a-like* [30].

Elegant work has identified that the transcription factors DUX, ZSCAN4, DPPA2 and DPPA4 activate 2C specific genes by binding to MT2 LTR promoters [30,31,32,33,34,35,36,37]. Mechanistically, MERVL LTRs, therefore, regulate genes by acting as poised promoters (Figure 2). Intriguingly, overexpression of the zinc finger protein, ZSCAN4 has also been described to protect cleavage embryos from DNA damage [38]. In Figure 3, we illustrate the MERVL-2C gene network by depicting DUX-targeted MT2 LTRs within 10 kb of 2C-associated genes [23,30]. The expression profile of *Zscan4c* mRNA as defined in [39] is also displayed (Figure 3, and Appendix A (Appendix A) for raw data). Of note, L1 elements also exert co-opted roles in totipotency and early developmental transitions [6,40,41]. Overexpression of the human DUX orthologue, DUX4 in human ESCs leads to an induction of ERVL promoters [35], which are usually expressed at the cleavage stage of human development [42]. Therefore, there are obvious parallels between regulation of mouse and human totipotency [26] and MERVL and ERVL are derived from the same retrovirus superfamily. ERV regulation of the 2C stage gene network serves as a striking example of convergent evolution [26] or convergent co-option, and further examples of convergent co-option are discussed below.

#### 2.1.2. Sequence-Specific Epigenetic Silencing

IAP elements (restricted to the ‘mus’ lineage, see Figure 1) are subject to sequence-specific epigenetic silencing through KRAB-zinc finger proteins (KZFPs), of which there are around 700 in the mouse genome [43]. KZFPs recruit KAP1 and SETDB1, which create heterochromatin foci that can spread into and repress neighbouring genes [21,44,45,46,47]. This concept is illustrated by the KZFP, ZFP932, which binds to a subfamily of IAP elements through a sequence within the proviral 3’ polypurine tract, which is a determinant of retroviral replication [48]. This regulatory sequence now serves to regulate expression of an IAP-proximal gene, *Bgalp3.* Inactivation of *Zfp932* results in loss of local silent chromatin marks and a gain of the enhancer marks H3K27ac and H3K4me1 and Pol II accumulation. Similarly, depletion of KAP1 or SETDB1 in mESCs or neural progenitor cells (NPCs) leads to multiple instances of increased expression of ERVs and their proximal genes [21,44,49]. This is accompanied by an epigenetic switch from a dual H3K9me3 and H4K20me3 repressed configuration to an enhancer signature, characterised by H3K27ac and H3K4me1 [21]. A causative role for IAP-embedded enhancers in regulating proximal genes upon KAP1-depletion was recently demonstrated, by employing strain-specific IAP-integrants [50].

Figure 4 provides an illustration of how IAP elements can exhibit genome-wide effects on gene expression through their epigenetic repression, using published data [21]: IAP elements are depicted that exhibit KAP1-dependent H3K9me3 peaks proximal to KAP1-regulated genes. It is not known if IAP elements exert a natural role in the tissue-specific regulation of these genes [21]. Mechanistically, IAP elements regulate genes through spreading heterochromatin from silencers and act as poised enhancers (see Figure 2). Recently, the KZFP, *Zfp708* has been discovered to exert transgenerational maintenance of DNA methylation at LTR retrotransposons [51]. Many KZFPs still have unknown roles, although the functions and binding profiles of clusters of young KZFPs have been assessed in a new study using knockout mice, coupled with chromatin-immunoprecipitation assays in mESCs [43].

#### 2.1.3. Metastable Epialleles

The best characterised examples of ERV integrants differentially regulating a gene between individual mice may be the *Agouti viable yellow* (*Avy*) [52,53] and *Axin-fused* (*Axin^Fu^*) [54] alleles, which arose due to insertions of IAP elements either upstream or within an intron of the *Agouti* or *Axin* genes, respectively. These IAPs are variably silenced by DNA methylation between individuals that are genetically identical at these loci, resulting in variable expression of *Agouti* and a range of coat colours, or expression of a truncated version of *Axin*, which causes a kinked tail phenotype. These and similar events, the majority of which are evolutionarily young, have been termed metastable epialleles and have been recently catalogued in a genome-wide screen, although only a few have been shown to alter gene expression thus far [18]. How these metastable epialleles arose and to what degree they function as regulators of gene expression remains unclear [55,56]. Importantly, many IAP copies are conserved across mouse strains and subject to KAP1/KZFP-mediated stable epigenetic repression (see above, Section 2.1.2) and the latter copies may play a more prominent role in repressing host genes than the polymorphic metastable epialleles, which may have arisen through mutation of cis-acting silencers, as has been documented to occur for L1 [57]. Future work on variably methylated IAP elements may further our understanding of genetic variation between individuals as well as providing insight into epigenetic silencing mechanisms of actively transposing ERVs.

#### 2.1.4. Position-Effect Variegation

Young IAP insertions can contribute to position-effect variegation (PEV). This is a phenomenon whereby genes or transgenes exhibit variegated expression in some cells but not others, due to their position nearby heterochromatin, which can spread. For example, it was shown that a strain-specific IAP insertion approximately 300 bp upstream of the *B3galtl* gene (beta 1,3-galactosyltransferase-like) could repress gene transcription through spreading of H3K9me3, H4K20me3 and DNA methylation, in mESCs [20]. Of note, the human silencing hub (HUSH) complex was identified as a novel epigenetic complex involved in PEV [58] and will be discussed below. Further examples of IAP elements regulating genes are being discovered regularly [59], suggesting that data thus far represent only the tip of the iceberg.

### 2.2. New TEs in Humans and Convergent Co-Option

The capacity for evolutionarily young intact TEs to repress proximal genes in development has also been documented in human models. For example, loss of the maintenance DNA methyltransferase, DNMT1 in human NPCs results in demethylation and transcription from young, hominoid-specific L1 antisense promoters, which can give rise to chimeric transcripts with proximal genes (<fifty kb away) [60]. This work builds on the previous discovery that the L1 5’UTR has an antisense promoter and kozak sequence and produces ORF0, which can form fusion proteins with proximal exons [61]. There are some similarities between DNMT1-depletion and KAP1-depletion but in the case of KAP1, upregulated genes were mainly proximal to HERVKs (HML2) and SVAs (of < seven myo and ~three myo, respectively) [62,63,64]. Of note, KAP1 can target the primer binding site (PBS) of HERVKs, in a similar way to its targeting the PBS of MLV [65,66,67], and this mechanism is known to silence an adjacent reporter promoter [63,68,69]. It is not known if the above TEs can naturally activate adjacent genes, since the above studies involve knockout of epigenetic modifiers.

An important example of a TE co-opted to activate genes is the hominoid-specific HERVH LTR7 [70] (Figure 1), which functions as an enhancer in pluripotent cells and is hypomethylated and expressed in differentiation-defective hIPSCs (human-induced pluripotent stem cells), reviewed in [7,71]. Similarly, upregulation of HERVK in pluripotent cells has been reported [64,72]. Notable studies have documented how TEs may regulate human genes in adult tissues, for example in CD4 + T cells and macrophages [68,73,74,75,76]. Still relatively little is known, however, about how the ever-evolving TE burden contributes to present-day human gene regulatory networks. By comparing gene expression and histone marks associated with functional and poised enhancers (H3K27ac and H3K4me1) across primate lineages [77,78], it has recently been shown that many regulatory regions are derived from new TE insertions (including SVA_B,C,D and F; LTR12 and Alu). For example, a human-specific SVA_F insertion located in the intron of the gene *Jarid2*, was identified to function as a silencer in the liver and nervous system [79].

Below, we will discuss several examples of convergent co-option. This will highlight how TEs exert parallel roles in the human and murine lineage, despite being species-specific, and emphasizes the need for more comparative genomics in future work.

#### 2.2.1. The HUSH Complex

The human silencing hub, or HUSH complex was identified in a screen for mediators of PEV in human cells [58,80] and is comprised of TASOR (also known as FAM208A), MPP8 (encoded by MPHOSPH8) and periphilin-1 (PPHLN1). HUSH is recruited to H3K9me3-dense genomic loci and partners with the chromatin remodeler, MORC2 [80,81,82] or MORC2A in mice [11]. The HUSH complex also partakes in the restriction of incoming exogenous retroviruses to which it is recruited through a novel DNA binding protein, NP220 (*ZNF638*) that is attracted to clusters of cytidines [83]. Although identified as a complex together with MPP8 and periphilin-1 in human cells, FAM208A was earlier identified as a novel epigenetic modifier in an ENU mutagenesis screen in mice [84]. FAM208A plays a critical role in development because homozygous mutant mice are not viable beyond gastrulation. We and others have shown that the HUSH complex is required to regulate expression of young (<five myo) transcriptionally active L1 elements (L1Md_F/A/T, see Figure 1) in mESCs [11,82,85]. It also represses genes that have accumulated these TEs upstream or within their introns, a trait shared by KAP1 [85]. Some of these genes are mouse-specific, suggesting they are recently evolved. The HUSH complex exerts a parallel role in human cells in regulating full-length L1 elements (L1PA4 and L1HS, see Figure 1) in the hominoidea lineage [82,86]. Similarly to mESCs, HUSH-regulated L1s are often located within introns of active genes, where they attract local H3K9me3, resulting in a slight downregulation of the genes in which they are positioned [82]. Genes repressed by the HUSH complex include KZFPs [58], which regulate TEs themselves. New data suggest that the HUSH complex targets RNA, revealing how it could be recruited to and exert epigenetic repression on transcriptionally active L1 elements [87].

#### 2.2.2. A TE Origin to Genomic Imprinting

Genomic imprinting refers to the differential DNA methylation at imprinting control regions (ICRs), established in the germline, which determines parental allele-specific expression of a set of imprinted genes [88]. This epigenetic mechanism occurs in eutherian mammals as well as, less frequently, in marsupials and is essential for the regulation of development. Historically postulated to be a phenomenon exemplifying “the battle of the sexes” [89], the evolutionary origin of genomic imprinting remains enigmatic. However, one prominent theory is that it arose from a DNA methylation-based defence mechanism against exogenous DNA [90]. There are various lines of evidence to support this theory reviewed in [91], including the fact that some, but not all, imprinted genes resemble TEs while others originate from retrotransposition events [92,93,94].

Two recent studies have shed light on the link between TEs and imprinting by demonstrating that species-specific ERVs epigenetically regulate mouse- and human-specific imprinted genes [95], as well as non-canonically imprinted genes in mouse extra-embryonic tissues [96]. Another link between imprinted genes and TEs relates to the role of KZFPs in the maintenance of genomic imprints: ZFP57 and the more recently identified ZNF445/ZFP445 [97] are both critical to this process in mice and humans and, interestingly, have both been shown to also bind to TEs [98,99], suggesting that the binding motif of these proteins may have derived from a TE. Likewise, a recent study into the stochastic loss of imprinting (LOI) between mouse ESC strains mapped this instability to a region of chromosome 13 that overlaps a cluster of KZFPs [100], including some which have been suggested to regulate sex-specific gene expression [101], further implicating KZFPs in the regulation of imprinted genes. Thus, while genomic imprinting is conserved in eutherian mammals, the contribution of TEs to the regulation of genomic imprinting is evolving in a species-specific manner.

#### 2.2.3. Fighting Fire with Fire: TEs as Effectors of Immunity

The innate immune system, while conserved among mammals, displays marked species-specificity in the transcriptional response to interferon signalling, consistent with its role in adaptation against pathogens [102,103,104]. MER41 is a primate-specific ERV, which has been shown to act as a poised enhancer (see Figure 1 and Figure 2) for a number of interferon-γ (IFNG)-stimulated genes through its recruitment of the transcription factor, STAT1 [24]. While the regulation of innate immunity genes by MER41 elements may be largely species-specific, MER41-like elements with the ability to act as IFN-dependent enhancers are common to many mammalian lineages [105]. Interestingly, in the mouse, which lacks any MER41-like elements, a murid-specific ERV RLTR30B is enriched for STAT1 binding and exhibits IFN-inducible enhancer activity in reporter assays [24]. The mammalian innate immune system, therefore, appears to have been recurrently but independently shaped in individual lineages by ERV-derived IFN-inducible enhancers.

## 3. Gene Regulation by Transposable Elements: The Old

The genomes of present-day species are rife with remnants of their previous bombardment with TE insertions throughout evolution. Such ancient elements have had ample time for mutations to render them incapable of mobilizing in the genome, while simultaneously evolving potentially beneficial roles. It has therefore been hypothesized that TEs that are conserved across species are more likely to have been co-opted [106]. Of note, the contribution of ancient TEs to gene regulatory networks is underestimated due to the erasure of ancient TEs through genetic drift and their loss over evolutionary time except for the transcription factor binding sites, which can be preserved under purifying selection.

### 3.1. When X-Chromosome Inactivation Is on the LINE

Perhaps the best illustration of the importance of gene regulation by TEs at a higher order chromatin level is X chromosome inactivation (XCI). In contrast to autosomal chromosomes, where cells inherit two copies of each chromosome, the sex chromosomes pose a problem of unequal dosage, whereby females receive two copies of the X chromosome, while males have only one. To resolve this problem an entire X chromosome is subject to transcriptional silencing in females during development. The mechanism behind this involves coating of the inactive X by the long non-coding RNA *Xist,* which is expressed from the inactive X chromosome [107], reviewed in [108].

Mary Lyon originally hypothesized that the high density of L1 elements across the X chromosome may facilitate XCI through heterochromatin spreading of H3K9me3 marks [109] and see Figure 2 for a diagram of heterochromatin spreading. Indeed, L1s occupy roughly twice as much of the X chromosome than they do of autosomal chromosomes [110] with an enrichment of L1M1 and L1P4 subclasses (see Figure 1). These elements were active during the transition between eutherians to prosimians, 60 to 100 million years ago. Interestingly, 10% of genes on the human X chromosome that escape inactivation are located in segments with significantly fewer L1s [110]. It was later discovered that the density of L1s is an important factor in enabling efficient XCI [111]. It remains to be determined if phase separation, which has been recently defined as a feature of heterochromatin spreading [112,113] applies to XCI. Intriguingly, there appears to be preferential invasion of the X chromosome by LINEs of any age [110], suggesting that these elements are continuously co-opted to play a role in XCI.

### 3.2. Ancient TEs Shaping the Brain of Mammals

Two remarkable examples that substantiate the Britten and Davidson hypothesis [114], wherein the emergence of novel structures or functions could be aided by the co-option of ancient TEs, involve structures specific to mammalian brains: the corpus callosum and the neocortex.

The corpus callosum: Tashiro and colleagues identified a Short INterspersed Element (SINE) locus to exert an enhancer function specifically in the corpus callosum [115]. This SINE locus (AS021) belonged to an ancient family of SINE elements conserved among amniotes with some copies over three hundred myo [116]. Using a *lacZ* reporter assay, SINE AS021 was shown to drive reporter expression specifically in mouse cortical neurons that project axons into the corpus callosum [117] and served as a natural enhancer for *Satb2,* a transcription factor (TF) involved in corpus callosum formation. Of note, the authors could identify several other conserved ‘Amniota SINE1s’ (AmnSINE1s, see Figure 1), with evidence of their co-option in regulation of the corpus callosum, which is interestingly only present in placental mammals [118]. The corpus callosum connects the two hemispheres of the brain, facilitating their communication.

The neocortex: another brain-specific enhancer derived from a TE is MER130 (MEdium Reiteration frequency), which is also conserved amongst amniotes (Figure 1). This enhancer was identified through mapping the genome-wide binding sites of the co-activator p300 in the developing mouse embryo and shown to be enriched in the neocortex of E14.5 mouse brains. Importantly, a number of MER130 elements were found to be marked with H3K27ac and to contain binding motifs for several TFs important for brain development. These MER130 loci were verified to function as enhancers using a *luciferase* assay and were located next to genes annotated as being associated with abnormal telencephalon morphology. Figure 5a shows the distribution of MER130 elements in the mouse genome and highlights those which are bound by p300 in the E14.5 neocortex as well as loci where this element overlaps a H3K27ac mark in the E14.5 whole brain. Here, we also reveal that (1) 54 of the 107 mouse MER130 elements described in this study are conserved in the human genome and (2) the 12 genes associated with abnormal telencephalon development in the mouse genome are also located in proximity to MER130s in the human genome. One example of this, *Zfp432/ZNF432*, is depicted in the figure. Furthermore, this subset of MER130s appear to overlap with DNase hypersensitivity sites in day 85 human brain (the equivalent timepoint to E14.5 in the mouse). These results illustrate that TEs have been co-opted as enhancers in the neocortex, another mammalian specific structure [119], and suggest that this function is conserved between mammalian lineages. The neocortex is involved in conscious thought and reasoning, and in humans it is involved in language. Notably, the neocortex has evolved a significantly different structure in primates compared to rodents with abundant folds to increase its overall surface area. Little is known about the origin of MER130 from which this enhancer is derived, except that it was likely once a DNA transposon.

### 3.3. Ancient Mammalian-Conserved TEs as Insulators

Co-opted repeats have also been documented from the ancient MIRs (mammalian-wide interspersed repeats), which are amongst the most ancient TE families in the human genome and are classified as tRNA-derived SINEs (see Figure 1). This repeat family was originally found to be relatively enriched in the proximity of transcriptional start sites and to correlate with tissue-specific gene expression [120]. Later analyses employing CD4+ T cells identified a small subset of MIRs (0.36%) that function as insulators, characterised by a presence of a B-box and their capacity to recruit Pol III. Their chromatin barrier activity appears to be cell-type specific for CD4+ T cells, where the authors could also identify differences in the expression levels of genes found on either side of the MIR-associated insulators [22]. Remarkably, a MIR integrant was proposed to have enabled differentiation of regulatory CD4+ T cells in placental mammals to reduce inflammatory immune responses that would otherwise target the foetus during pregnancy [121]. A similar example is the eutherian DNA transposase MER20 (see Figure 1), which possesses an insulator function suggested to have contributed to the process of differentiation of endometrial stromal cells by limiting the spread of heterochromatin [122]. See Figure 2 for a diagram of a TE function as an insulator.

## 4. Gene Regulation by Transposable Elements: The Ugly

Given the ability of TEs to regulate genes, including generating chimeric transcripts, it is not surprising that TE insertions can cause disease. The phenomenon whereby cancer progression imitates exaptation events occurring during evolution by employing TEs has been termed onco-exaptation [4]. In this section, we highlight examples whereby the epigenetic dysregulation of TEs has been shown to underlie human diseases. Examples of actively transposing TEs causing disease, although important will not be discussed here, since they are beyond the scope of this review.

### 4.1. Gene Dysregulation by TEs in Cancer

Like the aberrant activation of TEs following experimental disruption of epigenetic factors discussed above, in B-cell derived Hodgkin’s lymphoma, the de-repression of the simian MaLR-type LTR, THE1B (Figure 1) results in the expression of a non-canonical transcript of the proto-oncogene CSF1R. The derepressed LTR is silenced by DNA methylation in non-Hodgkin cell lines, whereas cells from these patients exhibit a loss of ETO2 (CBFA2T3), a transcriptional repressor interacting with histone deacetylases (HDACs). Additional transcripts originating from the THE subfamily of LTRs were also observed in Hodgkin’s lymphoma cells [123], illustrating that multiple derepressed LTRs can have further-reaching consequences in lymphoma development. In fact, a later study identified an additional derepressed LTR to result in the activation of interferon regulatory factor 5 (IRF5), a transcription factor previously shown to be necessary for the survival of Hodgkin Lymphoma cells [124]. Expression of IRF5 was driven by the primate-specific LTR, LOR1a, which was hypomethylated in samples showing aberrant expression of IRF5 [125]. Notably, it has also been reported that infection by Epstein-Barr Virus can lead to the activation of LTRs, such as ERV1 and ERVL, in the context of primary B-cells and lymphoblastoid cell lines [126]. New examples of LTRs functioning as enhancers to disease-relevant genes have recently been discovered for acute myeloid leukaemia [127]. Cancers are not only associated with upregulation of ERV regulatory elements but also the production of exapted TE-derived proteins that regulate the immune response [128].

### 4.2. Gene Dysregulation by TEs in Autoimmune Disease

As well as cancers, multiple instances of autoimmune disease, such as rheumatoid arthritis or multiple sclerosis involve some degree of aberrant TE activation [129]. In some such instances, natural or chimeric proteins produced by TEs when they are derepressed are thought to contribute to disease [130]. One example is in the case of Systemic Lupus Erythematosus (SLE), in which several TEs have been found to be hypomethylated in neutrophils from diseased patients, compared to healthy controls, particularly at L1 elements [131]. Importantly, genes upregulated in the neutrophils of SLE patients were found to be associated with the presence of L1s in the antisense orientation with respect to the gene. Upregulated genes with L1s were enriched for biological processes involving apoptosis and programmed cell death [132]. Remarkably, KAP1 and KZFPs have been linked to the pathogenesis of lupus, in which disease was associated with the expression of a retroviral envelope protein produced from an ERV that was under epigenetic regulation in healthy individuals through binding of several KZFPs to the ERV LTR [133]. Of note, TEs are emerging more broadly as key players in driving inflammation and autoimmunity, including neuroinflammation in the context of various brain disorders, as reviewed in [134,135].

The above examples represent a mere snapshot of the potential consequences of loss of epigenetic control of TEs. Despite little causative data in this area thus far, given the direct impact of the described TEs on genes with critical roles in these pathologies, it is possible that these events are causative rather than passenger events.

## 5. Conclusions

In this review, we have sought to illustrate how epigenetic repression of invading TEs has led to the evolution of epigenetic regulation of gene networks, into which invading TEs have become embedded. We have focused on evolutionary young (new) TEs, a fraction of which are still active, as well as old conserved TEs, drawing on mouse and human models, in order to capture some of the breadth of TE co-option into gene regulatory pathways. Figure 5 provides a summary of ancient (MER130) TE-driven conservation throughout placental mammals (Figure 5a), compared to young (IAP) TE-driven gene regulation that is not necessarily fixed across mouse strains (Figure 5b) and represents a unique snapshot of ongoing TE co-option in real time. We speculate that new TEs continually fulfil and are co-opted into the same roles as old TEs, as long as they outperform their predecessors in terms of acting as effective gene regulatory elements. We note that while work to date has identified many instances of individual TEs contributing to the regulation of host genes, very little is known about TE co-option more broadly, particularly into the potential regulation of whole gene expression programmes and biological systems. Since TEs can drive genome innovation and adaptation in response to pathogen challenge, we hypothesize that TEs will be uncovered to play a more prominent role in the evolution of the human immune system. We also highlight the benefit of cross-species comparative studies in future work on TE co-option.

Notable examples we have focused on of TEs regulating genes in this review are MERVL LTRs driving expression of early embryonic genes, IAP ERV silencers repressing proximal genes in mESCs, and TEs repeatedly co-opted to mediate genomic imprinting. Note that although we have limited our scope to mouse and human studies, exciting examples of gene regulation by TEs are widespread and extend to the control of fruit colour in the tomato plant, for example [136]. Co-option of TEs is not limited to discrete genes or networks and here we also discuss how TEs have contributed to the evolution of X chromosome inactivation, innate immunity and even to new brain structures such as the corpus callosum, which is involved in communication between left and right hemispheres of the brain. Finally, we discuss how the advantage of the host harnessing TEs as elegant tools for genome innovation also comes at a cost: TE integrants can also cause disease as exemplified by increased obesity and diabetes observed in yellow *Avy* mice, or cancers and autoimmune disease in humans.

## 6. Methods

### 6.1. Expression Analysis in Mouse Pre-Implantation Development

The annotation of MERVL LTRs (MT2_Mm) was extracted from the RepeatMasker track for mm10. Dux binding was obtained from [35], the intersection of peaks called in the two individual replicates was used to identify Dux-bound MT2s using [137]. A list of 2-cell expressed genes was compiled from [23,30] which was refined to only include genes that are within 10 kb downstream of Dux-bound LTRs. Fragments Per Kilobase of transcript per Million mapped reads (FPKMs) were calculated using data from [39] using [138,139,140]. Read counts for repeat families were calculated using [141] and normalised for library size. Dux expression was calculated by aligning reads to reference AM398147 with [139]. Mean values across replicates are reported.

### 6.2. Genome-Wide Visualization of Features

Data for KAP1 binding, location of H3K9me3 and gene expression values following KAP1-KO were taken from [21]. The annotation of IAP elements was extracted from the RepeatMasker track for mm9. Variability of repeat integrants was derived from [142]. IAP elements were considered variable when they overlapped a deletion in more than one of the inbred laboratory strains. Overlapping and closest features were calculated with [137] and visualized with [143]. MER130 annotation was obtained from [119] and converted to version mm10 of the mouse genome. p300 coordinates from E14.5 dorsal cerebral wall taken from [144], and converted to mm10. Broadpeaks for H3K27ac in E14.5 mouse whole brains were downloaded from the UCSC ENCODE [145] tracks. PhastCons [146] conservation scores calculated for a multiple alignment across 60 placental mammals species were retrieved from UCSC Table Browser [147] specifically for MER130 coordinates. Instances of MER130 were termed conserved when the conservation score was > 0.7 for more than half of the repeat’s length.

## Figures and Tables

**Figure 1 viruses-12-01089-f001:**
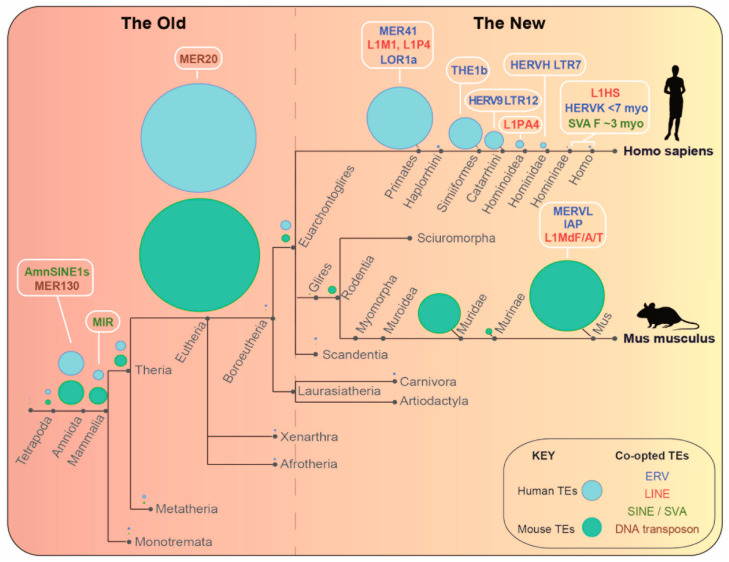
Evolutionary map of example co-opted transposable elements (TEs) for gene regulation. Clade diagram of human and mouse evolutionary trajectories overlaid with bubble plots showing the relative prevalence of all TEs in the human (blue) and mouse (green) genomes, according to their taxonomic specificity (from the *Dfam* database of repetitive DNA families). The dotted line represents an arbitrary evolutionary time cutoff to classify TEs as ‘Old’ vs. ‘New’ in this review. TEs discussed in this review are annotated within their respective taxa and colour coded according to their TE class (see the key). ERV; endogenous retrovirus, LINE; long interspersed elements; SINE; short interspersed elements, SVA; SINE/VNTR/Alu elements.

**Figure 2 viruses-12-01089-f002:**
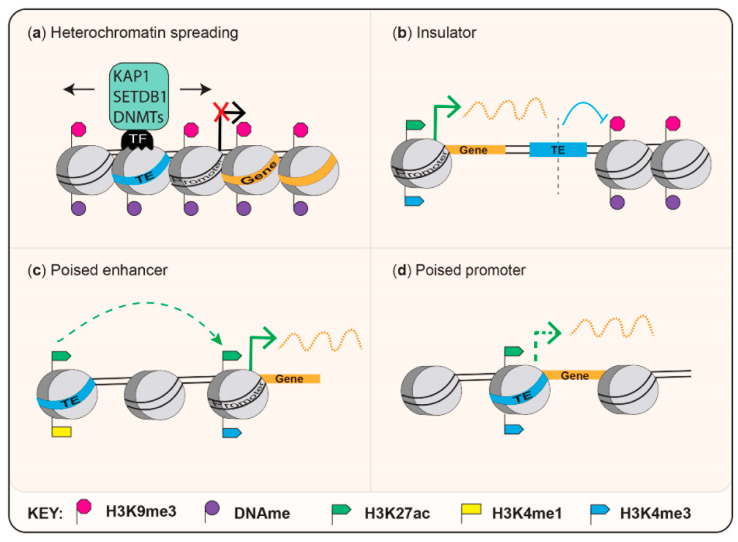
Mechanisms by which transposable elements (TEs) regulate host genes that are discussed in this review. Heterochromatin spreading into genes is represented in (**a**), which can occur stochastically but often radiates from silencers bound by sequence-specific transcription factor (TF) repressors. These can recruit heterochromatin-related proteins, some of which are highlighted here. Heterochromatin spreading has been described for IAP ERVs [20,21]. An insulator function is portrayed in (**b**), whereby a TE can protect a host gene from heterochromatin spreading and an example is MIR elements [22]. ‘Poised’ or cryptic TE-derived enhancers and promoters are shown in (**c**, **d**), which can be poised due to their dynamic epigenetic repression as discussed in this review. TE promoters can function as alternative promoters for protein-coding genes as is the case for MERVL long terminal repeats (LTRs), which can generate chimeric transcripts with host genes expressed at the 2-cell stage of mouse development [23]. MER41 has been shown to act as a poised enhancer [24].

**Figure 3 viruses-12-01089-f003:**
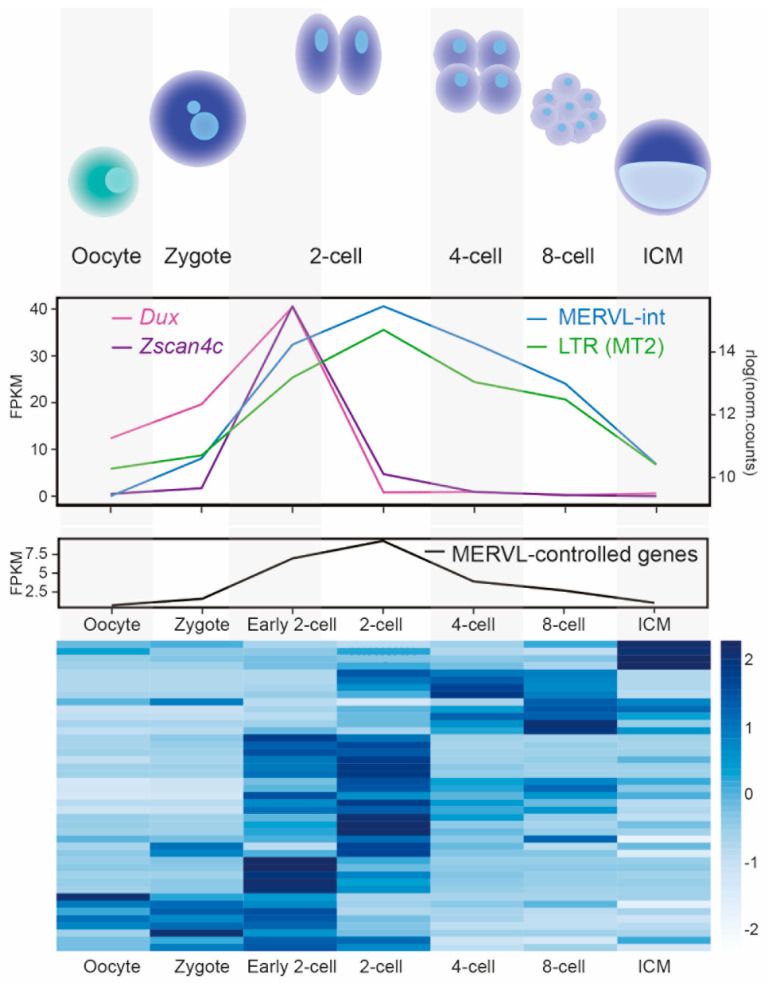
Temporal gene regulation through MERVL-derived LTR promoters. Top: Diagram of early mouse development. Middle: Median expression values (FPKM) of the transcription factors, *Dux* and *Zscan4c* (left axis) and of MERVL and its LTR promoter (MT2) as defined by ‘TEcounts’ software (right axis) are shown through development using data from [39]. Bottom: Expression of 43 2C stage-specific genes, controlled through MERVL LTRs, depicted both as the median expression in FPKM (’MERVL-controlled genes’ line graph) and as expression values of individual genes in a heatmap (values are scaled by row). A list of 2C stage-expressed genes was compiled by including previously published lists from both the following studies: [23,30]. FPKM expression values through development were extracted from [39]. Only genes that are within 10 kb downstream of MT2 LTRs that overlap a DUX binding peak (peaks extracted from [35]) and have detectable expression levels in [39] were considered here. See Appendix A (Appendix A) for raw data.

**Figure 4 viruses-12-01089-f004:**
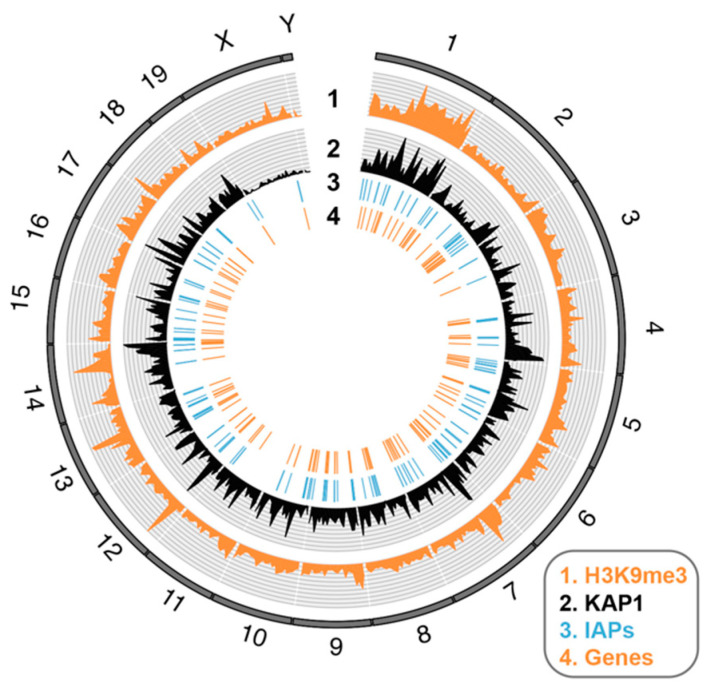
Heterochromatin spreading from silencers embedded in IAP-type ERV elements. Circos plot showing circular visualization of the mouse genome depicting IAP elements coated with silent chromatin marks and proximal genes that they potentially regulate. Data are from C57BL/6J-derived mESCs and extracted from [21]. Track 1 corresponds to a histogram of KAP1-dependent H3K9me3 coverage across 5-Mb windows. This represents regions of H3K9me3-enrichment that are present in KAP1 control mESCs but lost in KAP1 KO mESCs. Track 2 depicts a histogram of KAP1 peak coverage in 5-Mb windows. Track 3 shows individual occurrences of IAP elements (290) that overlap H3K9me3 peaks shown in track 1, and which are less than 10 kb from genes upregulated (>2 fold) in KAP1 KO mESCs. Note that KAP1-dependent H3K9me3 peaks are used rather than KAP1 peaks, due to their increased mappability at highly repetitive (young) IAPs, resulting from their spreading into genes. Track 4 depicts the coordinates of 145 unique genes <10 kb from IAPs in track 3 and upregulated in KAP1 KO mESCs. See Appendix A (Appendix A) for raw data.

**Figure 5 viruses-12-01089-f005:**
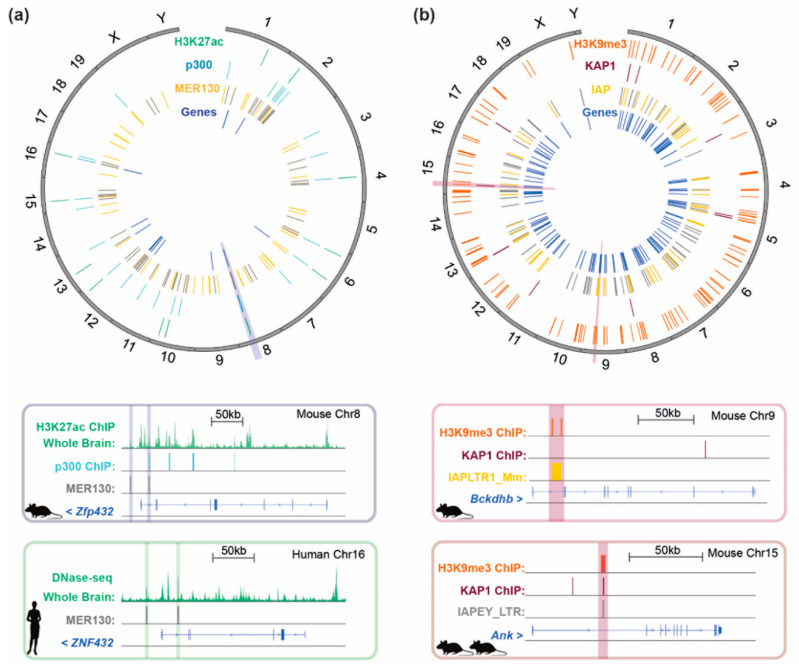
Chromatin landscape of past co-option events vs. co-option ‘in action’ of new TEs (**a**) Circos plot of MER130 TEs in the mouse genome according to [119]. (1) H3K27ac peaks (ENCODE) in E14.5 whole brain which overlap MER130 elements. (2) p300 binding sites in E14.5 dorsal cerebral wall, which overlap MER130 elements. (3) MER130 elements in the mouse genome depicted in yellow, except for those MER130 instances that are conserved in 60 placental mammals, including in the human genome, which are depicted in grey. (4) Genes located near to MER130 cortical enhancers which, when perturbed, result in abnormal telencephalon morphology [119]. (**b**) Circos plot of IAP elements in the mouse genome under KAP1 regulation according to [21]. (1) Coordinates of KAP1-dependent H3K9me3 peaks. (2) KAP1 peaks that overlap IAPs proximal (<10 kb) to genes upregulated upon KAP1 KO. (3) IAP elements which intersect H3K9me3 peaks in (1) and overlap or are proximal (<10 kb) to genes upregulated upon KAP1 KO. IAPs conserved between 14 inbred laboratory mouse strains are depicted in grey, whereas IAPs for which there is evidence of their absence in more than one strain (exhibiting strain variability) are highlighted in yellow. Genome browser windows in the lower half of the figure depict the highlighted TE/gene intersections from the circos plot in (**a**) showing conservation in the mouse and human genomes or (**b**) an IAP integrant that is only present in C57BL/6J-strain mice vs. an IAP integrant conserved across all 14 inbred mouse genomes. See Appendix A (Appendix A) for raw data.

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
