# Peer review of "Host Gene Regulation by Transposable Elements: The New, the Old and the Ugly"

_viruses, 2020, doi:10.3390/v12101089_

Round 1
Reviewer 2 Report
Enriquez-Gasca et al. provide an incisive summary of what is known about TE-derived epigenetic regulation in mouse and human. As a review, it was enjoyable to read and I appreciated demonstration of many points by example. The figures include new analyses, which add considerable value. While I formed a positive view of the work overall, I have a few suggestions for the authors to consider: 1) Broadly speaking, the authors focused on providing a summary of what is known. It could be useful to add in some places their thoughts on what is unknown, or what they would like to know. 2) Also along these lines, some aspects of the literature could be presented with a more critical eye, using the authors' expertise to guide readers as to what is missing or requiring improvement. For example, ref 18 (PMID: 30454646), a recent study from Ferguson-Smith, concluded that IAPs rarely act as epistable alleles, and here this is result is presented as "only a few have been shown to regulate gene expression thus far". This is arguably not a fully accurate representation of what that primary research concluded. 3) Line 69: "the mouse genome appears to contain active cohorts of ERVs as well as L1s [15,16]". Please provide a reference for mobile mouse L1s. 4) Line 79: "IAP elements are an abundant source of polymorphisms across mouse strains". In C3H mice yes, but in other strains it seems IAP is much less active (ref 59 from Rebollo, PMID: 32708087). Could soften this to say that IAPs are "a source of polymorphisms" but provide the most high profile examples in the literature. 5) Line 91: "Heterochromatin spreading has been described for IAP ERVs and L1 elements [20-22]." As far as I know, and please correct me if I'm wrong, heterochromatin applied to and spreading from new L1 insertions has not been demonstrated, instead the X-chromosome model cited here is one of L1 density acquired over time. If so, this should probably be made clear in this statement. 6) Line 202: The L1 antisense promoter has been shown to regulate numerous genes, including those expressed in the brain, by work prior to ref 62 (PMID: 31320637). It would be collegiate to cite Denli et al. (PMID: 26496605) or even go back further into the earlier papers from Speek and others. 7) Line 212: HERV-K is also strongly upregulated in hiPSCs (PMID: 26743714). 8) Line 213: "Few studies thus far have documented regulation of TEs and associated human genes in adult tissues". I agree that models of early development have had the most exploration, but I would argue that adult tissues have been published on extensively too, for examples see PMID: 30926969, PMID: 31822674, PMID: 19377475, PMID: 31230816, PMID: 29802231, PMID: 30381291, and others. The authors should pick which of these or other papers they cite but I would urge them to reconsider the statement. 9 Line 384: This section could also include reference to PMID: 30381291, which describes how EBV infection can activate LTRs as a prelude to lymphoma. Geoff Faulkner (University of Queensland)
Round 2
Reviewer 1 Report
I do accept the revision